# Biological Assessment of Zn–Based Absorbable Metals for Ureteral Stent Applications

**DOI:** 10.3390/ma12203325

**Published:** 2019-10-12

**Authors:** Devi Paramitha, Stéphane Chabaud, Stéphane Bolduc, Hendra Hermawan

**Affiliations:** 1Department of Mining, Metallurgical and Materials Engineering & CHU de Québec Research Center, Laval University, Quebec City, QC G1V 0A6, Canada; devi.paramitha.1@ulaval.ca; 2Centre de Recherche en Organogénèse Expérimentale/LOEX, Division of Regenerative Medicine, CHU de Québec Research Center, Laval University, Quebec City, QC G1J 1Z4, Canada; stephane.chabaud@crchudequebec.ulaval.ca

**Keywords:** absorbable metal, cytotoxicity, stent, ureteral, urothelial cells, zinc alloy

## Abstract

The use of ureteral stents to relieve urinary tract obstruction is still challenged by the problems of infection, encrustation, and compression, leading to the need for early removal procedures. Biodegradable ureteral stents, commonly made of polymers, have been proposed to overcome these problems. Recently, absorbable metals have been considered as potential materials offering both biodegradation and strength. This work proposed zinc-based absorbable metals by firstly evaluating their cytocompatibility toward normal primary human urothelial cells using 2D and 3D assays. In the 2D assay, the cells were exposed to different concentrations of metal extracts (i.e., 10 mg/mL of Zn–1Mg and 8.75 mg/mL of Zn–0.5Al) for up to 3 days and found that their cytoskeletal networks were affected but were recovered at day 3, as observed by immunofluorescence. In the 3D ureteral wall tissue construct, the cells formed a multilayered urothelium, as found in native tissue, with the presence of tight junctions at the superficial layer and laminin at the basal layer, indicating a healthy tissue condition even with the presence of the metal samples for up to 7 days of exposure. The basal cells attached to the metal surface as seen in a natural spreading state with pseudopodia and fusiform morphologies, indicating that the metals were non-toxic.

## 1. Introduction

The urinary tract is part of the renal system, and helps in maintenance of a homeostatic condition by draining the urine from the kidneys to the bladder and out of the body [1]. Obstruction to this tract can cause a build-up of urine leading to hydronephrosis or the swelling of a kidney, which later leads to the failure of other organ systems in the body [2]. Ureteral stents are widely used in patients with urological disorders when a relief of ureteral obstruction is needed, or when the maintenance of ureteral patency is required for healing purposes after ureteral and upper tract reconstruction, endoscopy, or trauma [3,4]. An ideal ureteral stent maintains excellent urine flow to optimize upper tract drainage, is resistant to encrustation, avoids infections, and is biodegradable [5]. The majority of urologists consider a routine placement of stents to be 1 to 6 weeks [6,7] following ureteral dilatation. This temporary need has led to the concept of biodegradable ureteral stents. Introduced in the late 1980s, biodegradable ureteral stent indication is dedicated for temporary treatment such as patients in line for surgery and the early-phase resolution of bladder outlet obstruction in patients waiting for the effect of a medical therapy [8]. Poly(glycolic acid), poly(lactic acid), and poly(lactic-co-glycolic acid) are among the biodegradable polymeric materials used for ureteral stents [5,9]. However, polymeric stents possess limited ability to resist external compression forces, such as those created by a malignant extrinsic ureteral obstruction, due to their intrinsic low radial strength, which is often reinforced by a metal skeleton made of non-degradable alloys [10,11]. At this point, metals that can degrade seem to constitute the ideal material for biodegradable ureteral stents. 

Nowadays, absorbable metals or metals that degrade gradually in vivo with an appropriate host response then dissolve completely upon assisting tissue healing [12] are available. Iron, magnesium, zinc, and their alloys are among the absorbable metals that have been studied, mostly for cardiovascular and orthopaedic applications. Coronary stents made of iron and its alloys have shown their safety and efficacy when tested in animals [13], while those made of magnesium alloys have been clinically tested for critical limb ischemia in human with encouraging results [14]. Meanwhile, zinc alloys have been more recently proposed as alternatives to iron and magnesium in view of their moderate corrosion rate, and their cytocompatibility has been proved toward various cell types [15,16,17]. In relation to ureteral stents, a preliminary study was done on the corrosion and antibacterial properties of Mg–Y alloy in artificial urine, suggesting the potential of absorbable magnesium in urological applications [18]. Since that preliminary description, studies on the potential use of absorbable magnesium for this application were reported. First, an in vitro and in vivo corrosion and compatibility study of Mg–6Zn was done in simulated body fluids (SBF) and a rat bladder model [19]. Another study introduced the use of human urothelial cells using three different culture methods with the presence of Mg–Y. The culture methods used were direct culture, where cells were seeded directly on the material; direct exposure culture, which was done by seeding the cells first and then putting the material on top of the cell layer; and exposure culture method, where cells and material were in the same environment and had interactions but were not in direct contact [20]. One study reported the biodegradation behavior of ZK60 in artificial urine and rat models. The results of the studies showed that absorbable magnesium has the potential to be used in urinary applications [21]. 

Zinc has been studied as a potential absorbable metal for vascular stent and bone implant applications. Zinc shows a moderate in vitro corrosion rate compared to iron and magnesium. The addition of magnesium has a strengthening effect on zinc; for instance, as-cast Zn–1–2 wt %) Mg possesses a tensile strength of 180 MPa, as high as that of most magnesium alloys, and with the absence of potential mutagenicity and genotoxicity [16]. In addition, an extruded Zn–1.2Mg showed a higher tensile strength of 360 MPa while maintaining an excellent ductility of 20%, and exhibited no potential toxicity [22]. Further alloying with aluminum also increases the mechanical properties of zinc; it has been shown that Zn–0.5Al and Zn–1Al possess tensile strengths of 203 MPa and 223 MPa, respectively, while maintaining high ductility, i.e., 33% and 25%, respectively [23]. Even though aluminum is considered to be potentially toxic, new, absorbable Zn–Al alloys are of interest for their use in the urinary system, where toxicity is less of a concern as it is implanted after the body’s natural filters, the liver and the kidney. Therefore, owing to their inherent strength and biodegradability, we propose zinc-based absorbable metals as potential materials for ureteral stents. This work aimed to assess the cytocompatibility of Zn–Mg and Zn–Al alloys towards human urothelial cells in 2D and 3D models.

## 2. Materials and Methods 

### 2.1. Materials and Specimen Preparation

This study began with a preliminary screening step involving a cell viability test and an electrochemical corrosion test on the three classes of absorbable metals: pure iron, magnesium, and zinc, before then focusing on selected zinc alloys. The first screening tests included a water soluble tetrazolium (WST) mitochondrial assay to assess the cell viability of urothelial cells treated for 3 days with the metal extracts, and a potentiodynamic polarization test to determine the corrosion rate in artificial urine solution, as detailed in another work [24]. The second screening test involved a further cell viability test with the addition of two zinc alloys (Zn–1Mg and Zn–0.5Al) and one magnesium alloy (Mg–2Zn–1Mn, or ZM21), which were prepared via casting and extrusion, as detailed in Mostaed et al. [23]. The alloys were turned into powders (particle size of 2–200 μm) by mechanical filing to be used for indirect 2D cytotoxicity, while small disks (surface area ~200 mm^2^) were cut to be used in the direct 3D cytotoxicity assay. Extracts of Zn–1Mg (10 mg/mL) and Zn–0.5Al (8.75 mg/mL) were prepared from the powders based on the 50% inhibition concentration (IC_50_) determined in the second cell viability test. The powders were sterilized under UV for at least 15 min and mixed with urothelial basal media: Dulbecco–Vogt modification of Eagle’s Medium (DMEM; Invitrogen, Burlington, Canada) and Ham’s F12 (Flow Laboratory, Mississauga, Canada) in a 3:1 proportion, supplemented with 10% fetal bovine serum (Hyclone, Logan, UT, USA), 5 µg/mL insulin (Sigma, St. Louis, MO, USA), 10 µg/mL epidermal growth factor (Austral Biologicals, San Ramon, CA, USA), 10^−10^ M cholera toxin (ICN, St. Laurent, Canada), 100 U/mL penicillin, and 0.4 µg/mL hydrocortisone (Calbiochem, San Diego, CA, USA). The mixture was then incubated in 8% CO_2_ at 37 °C for 72 h. After incubation, the solutions were filtered using 0.22 µm Durapore^®^ PVDF membrane filter (Millex^®^ GV, Merck Millipore, Ltd., Darmstadt, Germany) and stored at 4 °C prior to use. 

### 2.2. pH and Ion Measurement

The pH value of the metal extracts was measured using a pH meter (Beckman Coulter PHI 350; Beckman Coulter Life Sciences, Mississauga, Canada). In brief, the pH meter was calibrated using three pH standard solutions (pH 4, 7, and 10) at room temperature. First, the sample solutions were heated to 37 °C. They were then poured into a small beaker and measured repeatedly. The level of ions released into the extraction media was measured by inductive coupled plasma atomic emission spectrometer (ICP/OES, 5110 SVDV, Agilent Technology, Santa Clara, CA, USA) [25,26,27] for the three metal ions, i.e., zinc, magnesium, and aluminum. In brief, the extracts were digested for 3 days before the ion measurement. At day 1, 3 mL of extract was added into the vial, and a mark was drawn to show the meniscus level of 3 mL. In a fume hood, the crystallizer was filled with paraffin oil prior to vial holder mounting to avoid oil splash. The vial holder was then mounted, not touching the crystallizer bottom surface, and with the vial half immersed in the oil. The oil was heated to 90 °C and 3 mL HNO_3_ was added to the vial, upon which the temperature was raised to 115 °C. When the solution inside the vial reduced to 3 mL by observation of the mark, another 3 mL HNO_3_ was added. After the solution evaporated, leaving 3 mL inside the vial, the oil bath was lifted and air-dried. It was then placed on a paper towel to absorb the excess oil. The treatment at day 2 started with the addition of 600 μL nanopure water and 900 μL of 30% H_2_O_2_ into each vial. The oil bath bottle was then heated at 115 °C until the excitement phase appeared. Attention was paid to make sure that no drop of oil was lost by excessive excitement. The solutions were allowed to heat until reduced to approximately 3 mL. The oil bath was then lifted and air-dried and placed on a paper towel to absorb the excess oil. At day 3, the solutions were transferred into a 5 mL volumetric flask while the vial was rinsed with 3 mL nanopure water; the water was then poured into the volumetric flask to reach 5 mL. The solutions were then transferred into centrifuge tubes and ready to be analyzed. The solutions were then analyzed with the ICP/OES instrument.

### 2.3. Cell Extraction and Culture 

All procedures involving patients were conducted according to the Helsinki Declaration and were approved by the Research Ethical Committee of CHU de Québec-Université Laval (Project 1012–1341). Donors’ consent for tissue harvesting was obtained for each specimen and experimental procedures were performed in compliance with the CHU de Québec guidelines. Dermal biopsies were collected from the skin of healthy donors undergoing plastic surgeries. After extensive washes in phosphate-buffered solution (PBS) with 100 U/mL penicillin, 25 mg/mL gentamicin, and 0.5 mg/mL fungizone (Bristol-Myers Squibb, Montreal, Canada), the skin biopsy was cut into small pieces and incubated in Hepes buffer (pH 7.4) (MP Biomedicals, Montreal, Canada) containing 500 mg/mL thermolysin and 1 mM CaCl_2_ (Sigma, St. Louis, MO, USA) overnight at 4 °C. Following incubation, the epithelial layer was mechanically removed from the connective tissue containing fibroblasts (Fb). The dermis was transferred into a trypsination unit containing collagenase H solution (0.125 U/mL; Roche Diagnostics, Mississauga, Canada) in DMEM (Invitrogen, Burlington, Canada), 10% foetal bovine serum (Hyclone, Logan, UT, USA), 100 U/mL penicillin, 25 mg/mL gentamicin, and sodium bicarbonate (Fb media). After 4 h of incubation at 37 °C, cells were harvested by centrifugation at 300 *g* for 10 min. Fb were then seeded into culture flasks containing Fb medium and incubated at 37 °C in a humidified 8% CO_2_ atmosphere. Medium was changed three times a week. Normal human urothelial cells (NHUCs) were extracted from a healthy human urological tissue biopsy and were cultured as previously described [28,29]. NHUCs were seeded at a density of 5 × 10^5^ cells in 75 cm^2^ culture flasks with 1.5 × 10^5^ irradiated murine NIH/3T3 (ATCC^®^ CRL-1658™) as a feeder layer in urothelial basal media. Medium was changed three times a week. For 2D experiments, each experiment was done in five replicates.

### 2.4. Cell Viability Measurement in NHUC Monolayer Culture

Cells were seeded in 96 well plate at 10% confluence and were cultured in urothelial basal medium at 37 °C in a humidified 8% CO_2_ atmosphere. The next day (Time 0), WST-1 (Roche, Laval, Canada) was used following the manufacturer instructions. The stable tetrazolium salt in WST-1 is transformed into formazan by viable cells. Thus, the viability of cells correlates directly with the amount of formazan. Cell density was measured at the indicated time. Cell viability of metal-extract-treated cultures was determined in comparison to the cell viability in the untreated condition (100%).

### 2.5. Immunofluorescence for Cytoskeletal Observation 

NHUCs at a density of 1 × 10^5^ cells by cm^2^ were seeded in 24 well plates containing coverslips and incubated with 0.5 mL of metal extracts for 1 and 3 days at 37 °C with 8% CO_2_. The cells were fixed on chamber slides using cold methanol for 10 min at −20 °C, and then washed thoroughly with PBS. Detection of Keratin 8/18 allowed the visualization of cytoskeleton intermediate filaments whereas Hoechst staining allowed detection of nuclei. Next, 50 μL of guinea-pig anti-human keratin 8/18 antibody (ARP, Belmont, MA, USA) diluted 1:50 in PBS–BSA 1% was added and incubated for 45 min. After discarding the first antibody, the cells were rinsed using PBS. Next, 50 μL of Alexa Fluor^®^594 coupled with secondary antibody (A21203, Thermo Fisher Scientific, Waltham, MA, USA) diluted in PBS–BSA 1% was added, incubated for 30 min in the dark, and washed with PBS three times for 2 min each, and washed with distilled water twice. Next, 50 μL 1:100 dilution of Hoechst 33342 (Thermo Fisher Scientific, Waltham, MA, USA) diluted in PBS–BSA 1% was then added for 10 min of incubation in the dark and washed with distilled water three times. A drop of mounting medium, PBS–glycerol–gelatin (pH 7.6), was put on the slide, and the inverted coverslip was placed on top of the drop. The slides with coverslips were left at 4 °C overnight to make sure the mounting medium was solid. The slides were then viewed under a fluorescence microscope with ApoTome attachment (Zeiss-Axio Imager Z1, Toronto, Canada).

### 2.6. Flat 3D Ureteral Wall Model Preparation

A flat, 3D construct tissue model (a patch-like tissue) was prepared by adapting our established engineered bladder and ureteral tissue model using the self-assembly method [30,31]. The procedure was adapted from the bladder model as described by Chabaud et al. [32], and characterized by Bureau et al. [33], and Goulet et al. [34]. Normal primary human skin Fb cells were seeded at 3 × 10^4^ cells/cm^2^ in 6 well plates. An anchorage paper device had been previously placed in each well as a stroma support system. The cells were cultured in DMEM supplemented with 10% newborn calf serum (NBCS), 100 U/mL penicillin, 25 μg/mL gentamycin (Fb medium), and fresh 50 μg/mL ascorbic acid, and were incubated in 8% CO_2_ at 37 °C. The media was changed every two days. After 14 days, a second seeding of Fb took place and the culture was pursued for 14 additional days until stroma sheets were formed. Further steps were done to build a flat 3D ureteral wall (UW) tissue model (hence named “UW model”). NHUCs were seeded at 2 × 10^5^ cells/cm^2^ on each stroma sheet in NHUC basal media, supplemented with fresh 50 μg/mL ascorbic acid and incubated in 8% CO_2_ at 37 °C for 7 days under submerged conditions. The stroma sheets were then moved into petri dishes to provide elevation at the air–liquid interface for 21 days. The provided interface aimed for the differentiation and maturation of NHUCs into the urothelium layer and formed the UW model. 

A direct toxicity test was done by putting metal disks on the top of mature urothelium of the UW model (Figure 1). This simple method mimics the in vivo conditions, and was the first tissue-engineered model used to assess absorbable metal toxicity. The test assessed the cell function in the presence of the metal disk and its corrosion products, and also allowed observation of the effect of compression from the weight of the disk on the tissue, simulating the compression of a stent in the ureter. Disks of ZM21 (355 ± 0.02 mg), Zn–1Mg (801 ± 0.03 mg), and stainless steel 316L (437 ± 0.02 mg control) were used. The alloys were expected to show different toxicities in the 3D culture setting compared to the 2D. 

### 2.7. 3D Cell Function Evaluation

Direct toxicity tests were conducted to assess the urothelial cell function. Metal disks were sterilized with UV light exposure for a minimum of 15 min on each side, then dipped in ethanol three times for 5 s each and washed with sterile PBS under a sterilized atmosphere. The disks were put on top of the UW model and incubated for 1 week. The UW tissues were then sacrificed by cutting them into two halves. The first half was fixed with Histochoice tissue fixative (Amresco, Solon, OH, USA) and then embedded in paraffin. Histological sections were made by cutting paraffinized tissue into 5 μm size and staining with Masson’s Trichrome. The second half was embedded in frozen tissue medium (optimal cut temperature (OCT) compound; Tissue-Tek, Bayer, Etobicoke, Canada) and then cut into 5 μm for further immunofluorescence (IF) evaluation. The slides from both paraffinized (IF against uroplakin) and OCT (other IFs) were fixed in cold 100% methanol, blocked with PBS–BSA 1%, and incubated with primary antibodies, as described above in the procedure for cytoskeletal observation. Primary antibodies used were ZO-1 for tight junction (1:50; 40–2200, Thermo Fisher Scientific, Waltham, MA, USA), Ki-67 for cell proliferation (1:400; ab15580, Abcam, Cambridge, MA, USA), and laminin-5 for basement membrane (1:400; ab14509, Abcam, Cambridge, MA, USA). Secondary antibodies were coupled with Alexa Fluor^®^ dyes (Thermo Fisher Scientific, Waltham, MA, USA): Alexa Fluor^®^488 (1:500; A21202) and 594 (1:200; A21468). The slides were then viewed under the fluorescence microscope [30]. 

### 2.8. Scanning Electron Microscopy

Metal disk remnants were preserved in 2.5% glutaraldehyde overnight in 4 °C and processed for further observation under electron microscopy, as described by Heckman et al. [35]. After overnight incubation, remnants were washed three times with PBS for 5 min each and then dehydrated by using serial ascending concentrations ethanol solutions from 30%, 50%, 70%, to 90% for 5 min each, and then 3 × 5 min in 100% ethanol. The remnants were dried and then gold sputtered. Observation was done using a scanning electron microscope (JEOL 7500-F, JEOL Ltd., Tokyo, Japan).

### 2.9. Statistical Analysis

All results are expressed as mean ± standard deviation of each independent experiment. The statistically significant differences between the mean were calculated by ANOVA and post hoc Tukey’s test for multiple comparisons, with the level of significance selected at p < 0.05 using SPSS 25 (IBM Canada Ltd., Markham, Canada).

## 3. Results and Discussion

### 3.1. Screening Tests

The cell viability test revealed that the viability of the urothelial cells exposed to pure iron was lower than in those exposed to pure magnesium and pure zinc (Figure 2a). The viability of the urothelial cells exposed to pure magnesium increased with the metal’s incubation time. The viability of urothelial cells exposed to pure zinc remained high for all incubation times (Figure 2a). In artificial urine solution, magnesium corroded faster than zinc and iron (Figure 2b), confirming the previously published results [24]. The constant trend of the cell viability exposed to zinc could be related to the constant release of Zn^2+^ ions into the solution throughout the metal incubation period. Meanwhile, the low cell viability results of pure iron could be related to the formation of iron free radicals, causing oxidative stress. Free radicals react with organic molecules in the cell membrane, initiate a lipid-peroxidation process and eventually lead to cell death [36]. The increasing viability trend of cells exposed to pure magnesium could be related to the formation of magnesium salts, such as Mg(OH)_2_, Mg_3_(PO4)_2_, MgCO_3_, Ca(OH)_2_, Ca_3_(PO_4_)_2_, and CaCO_3_, that in turn reduced the availability of Mg^2+^ ions to interact with the cells [37]. The environment was considered unfavorable for the cells when the corrosion products, i.e., Mg^2+^ and OH^−^, reached a certain concentration [20]. From these results, iron was eliminated from further tests because of its low corrosion rate and its high toxicity toward urothelial cells.

In the second part of screening test, pure magnesium, pure zinc, Zn–1Mg, and Zn–0.5Al were used. All metal extracts induced an alkalinization of the urothelium basal medium, increasing its pH value from 8.05 to 8.34 for Zn–0.5Al, 8.45 for pure zinc, 8.50 for Zn–1Mg, and the highest value, 9.18, for pure magnesium (Figure 2c). The pH increase was mainly caused by the production of OH^−^ ions during the corrosion of those metals [38]. The faster the corrosion, the more OH^−^ was released and the more rapid the alkalinization, until OH^−^ reached saturation and the pH value became stable [19]. A pH increase influences cellular function and compatibility [37]. Moreover, the viability of urothelial cells decreased as the metal extract concentration increased (Figure 2d), a common condition for many type of cells [15,16,22,39]. Tian et al. [20] reported that starting from pH 8.3, urothelial cells were seen to be unhealthy, with round morphology, and when pH values reached 8.6 the reduction in the density of urothelial cells was significant. Another study on fibroblasts and keratinocytes found that these cells proliferated and migrated better and were more viable in pH 8–8.7 compared to acidic or more alkaline environments [40]. Gu et al. [41] reported that alkaline stress causes severe cytotoxicity in human osteosarcoma cells MG63. Similar results have also been shown for osteoblasts, but the cells have better performance in terms of gene expression and mineralization [42]. Urine has a wide pH range, from 4.5 to 8.0 depending on the body’s acid–base equilibrium state. On a regular, average diet, the pH range decreases to 5.0 to 6.5. Consistently acidic urine, i.e., pH ≤ 6.0, has been associated with an increased risk of bladder cancer [43]. Urine pH mainly depends on the bicarbonate (HCO_3_^−^) concentration in blood, where the higher the concentration, the higher the pH is [44]. Other substances in urine include inorganic cations (sodium, potassium, ammonium, calcium, and magnesium) and anions (chloride, sulphate, and phosphate), and organic components such as urate, creatinine, oxalate, and citrate [45].

By plotting the cell viability versus metal extract concentration, the IC_50_ values were determined as the following: pure magnesium (2.5 mg/mL), Zn–1Mg (10 mg/mL), pure zinc (8 mg/mL), and Zn–0.5Al (8.75 mg/mL). Based on this result, Zn alloys were chosen for further biological assessment. The concentrations of metal ions in 10 mg/mL of Zn–1Mg and 8.75 mg/mL of Zn–0.5Al after 72 h of incubation in urothelial basal medium are presented in Table 1. The detected magnesium ion in the Zn–0.5Al extract ca,e from the magnesium content in urothelial basal medium.

Murni et al. [15] reported that 0.49 ppm of zinc ions in a Zn–3Mg alloy extract killed 50% of a normal human osteoblast cell population. Kubasek et al. [16] found that the highest safe concentration of zinc ions for U-2 osteosarcoma and L929 fibroblast cell lines was 7.85 ppm and 5.23 ppm, respectively. In the present study, 25.96 ppm of zinc ions killed half of the urothelial cell population. Different cell line behave differently; one type of cell can be more sensitive to zinc ions than others. Urothelial cells, on the other hand, only present a phenotype similar to the basal cells when cultivated in monolayer culture, which is found in the deepest layer of urothelium in the native tissue. This caused the sensitivity of urothelial cells toward the metal extracts to be higher in the present study. In addition, the urothelial cells used in this study were in passage-2 and were harvested from a primary culture. Generally, primary cultures are more sensitive toward toxic substances than immortal cell lines, because the cell culture induces an adaptation of the cells to the culture conditions [46]. The use of serum (FBS) in the metal extracts or salt solutions gave a protective effect to the cells. It is well known that the use of serum in culture media at either 5% or 10% induces the growth of cells, and studies have reported that cell viability in cytotoxicity tests is higher in tests done with the presence of serum [16,47,48].

### 3.2. Cytoskeletal Observation

Some changes occurred in the cell size and morphology of NHUCs in the presence of metal ions (Figure 3). These cells still have the capacity to form colonies even when they are experiencing changes, but cells in the Zn–1Mg group shrank and appeared rounded, as in apoptotic stage, compared to Zn–0.5Al group at day 1 (Figure 3a–c). The intensity of cytokeratin (CK) was less in the metal alloy groups compared to controls. Observation at day 3 showed that cells in the Zn–1Mg group survived and those in the Zn–0.5Al group had recovered their normal state (Figure 3d–f). The cells in the Zn–1Mg group did not fully recover, as seen from their size and morphology compared to controls, and the intensity of their CK was also less than the other groups. Rounded morphology found in Zn–1Mg group at day 1 could also have been a result of the alkaline pH (8.50) of the culture media.

Cytoskeletal changes in the presence of Zn ions were reported by Murni et al. It was observed that the cells underwent changes in size, contour, and skeletal intensity, which corresponded to cell stress and inhibition of cell proliferation. The adaptation of epithelium to changing conditions is usually accompanied by transition in the cytoskeleton of epithelial cells. Cell stress leads to apoptosis events. CK, as a major family of structural protein in epithelial cells, plays a role in mechanical and non-mechanical functions, including protection from cell stress and apoptosis. Meanwhile, epithelial polarity helps in cell–cell adhesion and also in attachment of the epithelial cells to the underlining connective tissue. Besides this structural function, CK also play a role in dynamic processes such as mitosis, mobility, and differentiation [49,50]. CK 8/18 pairs are obligate partners and establish the primary CK pair in many epithelia, including urothelium [51]. CK’s roles in apoptosis have been reported by many studies. The apoptosis process, with chromatin condensation as its hallmark, is first preceded by the breakdown of CK8/18. The breakdown results in the collapse of the cytoplasmic and nuclear cytoskeleton [49].

### 3.3. Cell Function Evaluation

The UW model offered several very interesting characteristics, such as the absence of exogenous material use during the reconstruction, which allowed differentiation of the urothelium close to what happens in native tissue. This adequate urothelium maturation was confirmed by the water-tightness of the 3D construct, similar to that of the native porcine bladder [33,34]. Since urothelium is a highly specialized epithelium, the use of UW wall with an organization close to the native tissue, in combination with direct toxicity, was considered to be a simple method with which to represent in vivo conditions. Table 2 presents the changes observed macroscopically during the incubation of the UW models with the metal disks. The ZM21 group induced significant changes compared to the control and Zn–1Mg groups, as marked by changes in all three parameters.

Histological sections stained by Masson’s Trichrome provided information about the morphology of urothelium in the presence of the metal disks and their corrosion products. The tissue-engineered ureter used in this study consisted of layers of urothelium and stroma (connective tissue) beneath it, which closely mimicked native ureteral tissue. The ability of the urothelial cells to migrate, proliferate, and differentiate to form urothelium is important for maintenance of urothelial integrity and retaining the permeability barrier function [52]. Figure 1d shows the division of the tissue into four areas, from the most proximal area to the metal disk to the most distal one. In all groups, including the control group, which was a disk of the surgical grade stainless steel, in the section stained from area 1 (in direct contact with or under metal disks) no cells or dead cells could be seen (Figure 4a,e,i). In area 2, the layers of urothelium seemed damaged in the control and Zn–1Mg groups, (respectively Figure 4b,j), or consisted of a thin epithelium in the case of the ZM21 group (Figure 4f). The transition from the absence of urothelium to a thin damaged epithelium was gradual, as well depicted in Figure 4j. In area 3, the urothelium seemed unaffected, and the superficial urothelial cells were distinct in the control (Figure 4c) and the Zn–1Mg group (Figure 4k), whereas they formed a wave-like structure in the ZM21 group (Figure 4g). This area is known to be the site of bubbles and thick white structures; it seems obvious that bubbles pushed the urothelium into this shape. The bubbles were made of hydrogen as a result of magnesium corrosion, which can be written as an overall reaction of: Mg + 2H_2_O → Mg(OH)_2_ + H_2_, where the Mg(OH)_2_ further reacts with Cl^−^ ions to form soluble MgCl_2_ [38]. Finally, in area 4, the urothelium remained unaffected by the metal disks, whatever the condition. Based on the Masson’s Trichrome staining results, the metal alloy groups showed similar morphology compared to the controls.

Differentiated urothelial cells, also known as umbrella cells, assume their role in maintaining the barrier function by producing specific proteins such as zonula occludens (ZO)-1. Other proteins, such as Ki67, allow evaluation of the proliferative state of the tissue, which could be overexpressed in the case of a tissue physiological or pathological response (e.g., injury or cancer). Laminin-5 is also an important molecule because its presence in the basal lamina is an absolute requirement for adequate urothelial cell differentiation, as evidenced by Rousseau et al. [53]. Figure 5 presents the ZO-1, Ki67, and laminin-5 immunofluorescence staining of the histological sections of the UW model. ZO-1 staining was done to evaluate the presence of tight junctions between urothelial cells. Figure 5a–c shows tight junctions separating the superficial urothelial cells in each group. These tight junctions, together with uroplakin, play an important role in urothelium, acting as a barrier from urine leakage, ions, or any other compound in the urine. Cell proliferation processes were not observed for the UW model, as shown by the negative result of the Ki67 staining (Figure 5d–f). Laminin-5 staining confirmed that delamination or detachment of basal layer did not occur in either the control or the metal alloy groups (Figure 5g–i).

Tight junctions (TJ) between umbrella cells represent the main permeability barrier in combination with another protein, uroplakins [54]. TJs restrict paracellular diffusion and movement, and they contribute to maintenance of the surface polarity of cells by restricting the movement of proteins and lipids between membrane compartments. TJs are composed of cytoplasmic proteins, such as ZO-1, linking TJs to the cytoskeleton, and integral transmembrane proteins, such as occludin, junctional adhesion molecule, and claudins [55,56,57].

Laminins are a group of proteins that are the most important components of the basal membrane. Laminins play a role in the interaction between urothelial cells and the extracellular matrix. In the ureteral mucosa, laminins are expressed by the basal cells and support the assembly of the basal membrane, contributing to maintenance of cell and tissue integrity. Laminins are also involved in cell adhesion, hemidesmosome formation, migration, proliferation, differentiation, and prevention of programmed cell death [58,59,60,61]. The presence of two important proteins, ZO-1 and laminin-5, was noted in our UW model with the direct toxicity method. This means that the urothelium function of producing barrier proteins was not interrupted by the metals and their corrosion products (except in the area directly under the disk, where the epithelium was destroyed). The urothelium may also have preserved its permeable barrier function in close proximity to the cells.

### 3.4. Scanning Electron Microscopy Analysis

Evidence of corrosion of metals was observed in alloy groups after 7 days of incubation, while the control group showed the same surface as before the incubation (Figure 6a,b). Corrosion of ZM21 was seen as layers of metal flakes all over the surface (Figure 6d,e), while the Zn–1Mg sample had micro-size holes with attached precipitation on some parts of its surface (Figure 6g,h). Urothelial cells and stroma were observed on the surface of each metal disk (Figure 6c,f,i). Attachment of cells is marked by pseudopodia structures in cells with flattened shape. These results confirmed that the lost or removed parts of urothelium from area 1 in the UW model were actually attached to the metal and were removed together with the metal disks. The cells that were attached 3–13 μm in size, indicating that they were basal and intermediate cells [62]. This indicates that the compression of the urothelium probably inhibited the urothelial cells from differentiating well, which occurred similarly in the control and alloy groups. Basal and intermediate cells alone, i.e., non-differentiated urothelial cells, do not have the ability to produce such proteins able to maintain the impermeable barrier. These very encouraging results pave the way for further studies using a more representative metal structure, e.g., a metal mesh, to closely simulate the ureteral stent.

## 4. Conclusions

To our knowledge, this is the first study reporting a biological assessment of absorbable metals in view of ureteral stent application using normal human urothelial cells (NHUCs) and a metal direct toxicity test on a flat 3D ureteral wall tissue model that mimics the in vivo conditions. The NHUCs exhibited a survival mode in response to the toxicity of the magnesium and zinc extracts in monolayer culture, shown by reduced viability and unhealthy appearance, but recovered after some days. This result suggests that the cells could rapidly colonize damaged areas to restore mature urothelium, restoring the barrier function to block urine extravasation in the body. The direct toxicity test in the 3D ureteral wall model showed that the corrosion of metals had an effect on the morphology of cell layers, but the urothelium was still able to maintain its function in producing the barrier protein and may have had a permeable barrier function in close proximity to the cells. This indicates that the tissue was healthy despite the presence of the metal disks and their corrosion products after 7 days of exposure. The attached cell layers on the surface of metal disks may indicate that the metal specimens were non-toxic. The zinc alloy group seemed to be more cytocompatible toward human urothelial cells compared to magnesium alloys, and could be further exploited for biodegradable ureteral stent applications. In the future, in vitro test using a flat 3D ureteral wall tissue model could be used to partially replace animal testing in the initial steps of optimization of absorbable metal stents.

## Figures and Tables

**Figure 1 materials-12-03325-f001:**
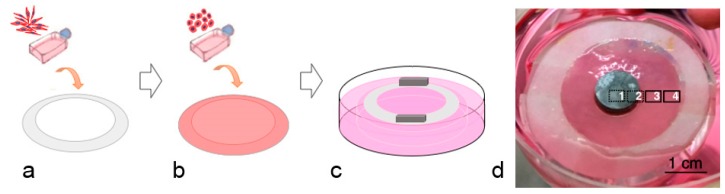
The ureteral wall (UW) model preparation: (**a**) production of stroma, 28 days culture, (**b**) production of tissue equivalent, 7 days culture in submerged condition, (**c**) displacement of tissue equivalent into petri dish, 21 days culture, (**d**) macroscopic view of the UW model with metal in place for direct toxicity test, the areas marked 1–4 are the area used for histological analysis.

**Figure 2 materials-12-03325-f002:**
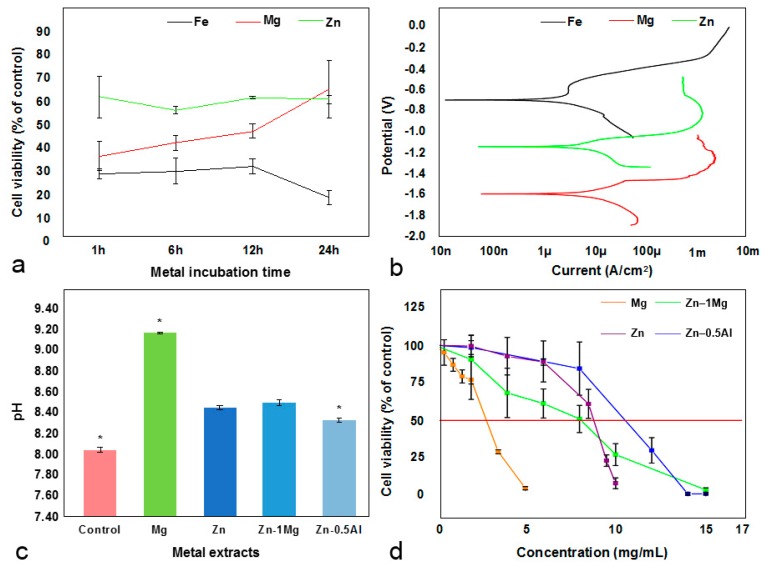
Screening test results: (**a**) viability of urothelial cells as a function of the metal incubation time in artificial urine solution, (**b**) potentiodynamic polarization curves of pure iron, magnesium, and zinc (potential in Volt (V) vs. saturated calomel electrode (SCE)), (**c**) pH value of metal extracts in urothelial basal medium, * *p* < 0.01, (**d**) viability of urothelial cells treated with different concentrations of metal extracts. Red line marks 50% of cell viability.

**Figure 3 materials-12-03325-f003:**
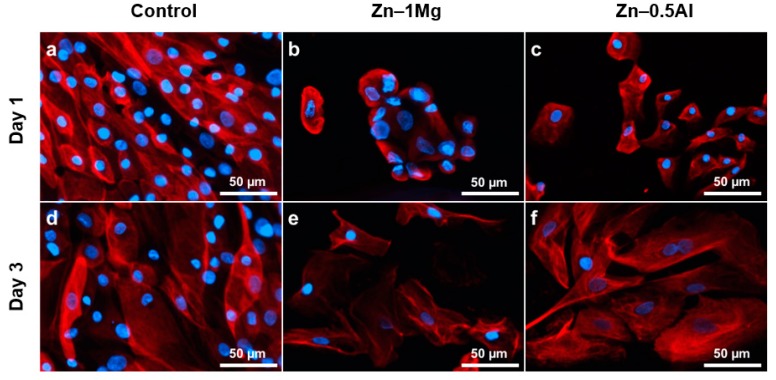
Images of cytoskeletal observation at days 1 and 3 in controls, Zn–1Mg, and Zn–0.5Al groups. Note: anti-cytokeratin 8/18 (red) staining was used to examine the changes of keratin, and Hoechst (blue) was used to stain nucleus of the normal human urothelial cells (NHUCs).

**Figure 4 materials-12-03325-f004:**
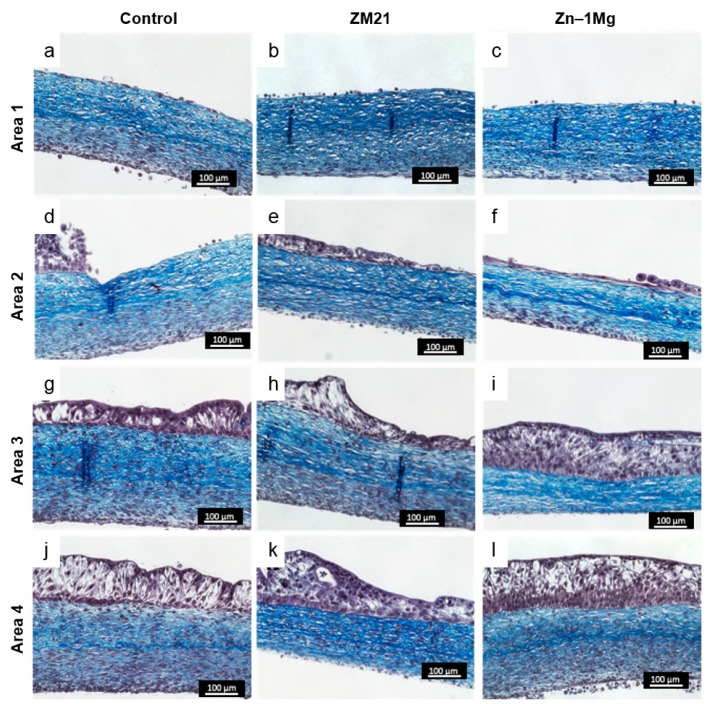
Histological analysis of UW tissue with different metal disks, stained with Masson’s Trichrome. Nucleus stained as purple and stroma/collagen stained as blue. Tissue sections are divided into four areas: (**1**) under/in contact with metal disks, (**2**) peri-metal site, (**3**) nearest area to peri-metal site, and (**4**) farthest area from peri-metal site. Bar = 100 μm.

**Figure 5 materials-12-03325-f005:**
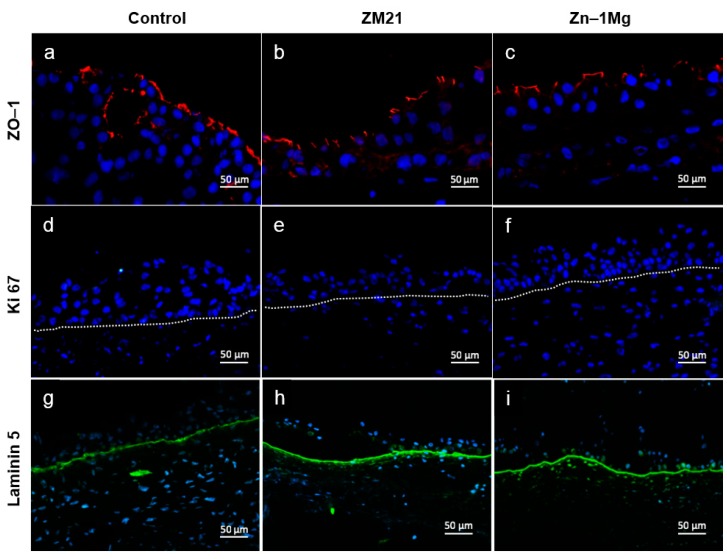
Expression of urothelial differentiation-associated protein and basal lamina in response to contact with metal disk after 7 days of incubation: (**a**–**c**) immunofluorescence anti-ZO-1 (red), (**d**–**f**) Ki67 (green), and (**g**–**i**) anti-laminin-5 (green). Nuclei were stained with Hoechst (blue).

**Figure 6 materials-12-03325-f006:**
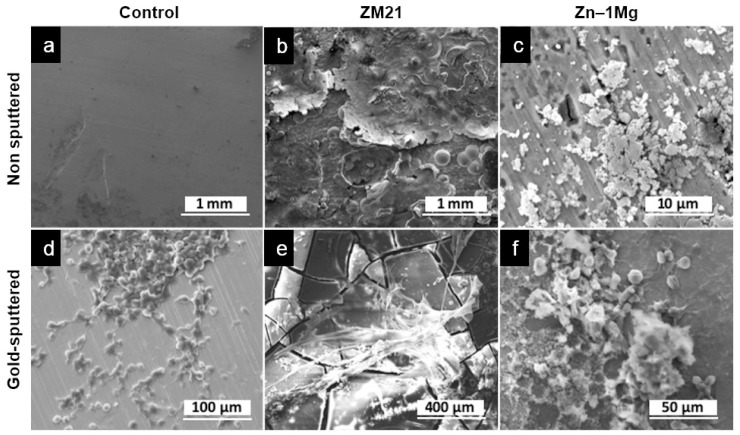
Scanning electron microscope micrographs of the surface of metal disks incubated for 7 days on the UW model: (**a**–**c**) corrosion layer was observed on the alloys’ surface, while the control showed no change, (**d**–**f**), urothelial cells and stroma collagen layer attachments were observed on the surface of each disk.

**Table 1 materials-12-03325-t001:** Ion concentration of metal alloy extracts at IC_50_.

Metal	Ion Concentration (ppm)		
	Mg	Zn	Al
Zn–1Mg	21.85–30.10	18.26–25.96	0
Zn–0.5Al	21.47–28.59	17.65–20.02	0.030–0.090

**Table 2 materials-12-03325-t002:** Macroscopic changes on the UW model observed after 1 week.

Parameters	Control	ZM21	Zn–1Mg
Tissue appearance	No change	Thick white structure surrounding the peri-metal site	No change
Medium culture (phenol red as pH indicator)	Yellow after 2 days of culture (acidic)	No color change (alkaline)	Yellow after 2 days of culture (acidic)
Metal corrosion product	Not found	Bubbles surrounding the metal	Not found

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
