# Peer review of "Biological Assessment of Zn–Based Absorbable Metals for Ureteral Stent Applications"

_materials, 2019, doi:10.3390/ma12203325_

Round 1

Reviewer 1 Report

Absorbable metals or metallic stents, metals that corrode in physiological environment, constitute a new class of biomaterials intended for temporary medical implant applications. The results presented in the manuscript are new and interesting since Absorbable metals are suggested to use for ureteral stent applications. Moreover, new absorbable Zn-Al alloy also is suggested to use for this purposes. However some improvement of the manuscript should be done:

Introduction part. Besides of the Review article of H. Hermawan [12], two additional important Reviews also should be cited and discussed in the manuscript: (1) Recent Advances in Biodegradable Metals for Medical Sutures: A Critical Review; by J.M. Seitz et al. ADVANCED HEALTHCARE MATERIALS 4 (2015) 1915-1936 and (2) Biodegradable Metals for Cardiovascular Stents: from Clinical Concerns to Recent Zn-Alloys; by P.K. Bowen et al. ADVANCED HEALTHCARE MATERIALS 5 (2016) 1121-1140. Figure 2 a and b. The data regarding Fe viability of urothelial cell and potentiodynamic polarization curve should be removed since they are redundant in this manuscript. Figure 6. The beginning of the caption for figure “Scanning electron microscope images...“ should be changed to “SEM micrographs…”.

The manuscript could be accepted for publication in Materials after minor revision.

Author Response

We thank you very much for the time taken by the reviewers to review our manuscript. We are very grateful for the comments and carefully responded to them as detailed below.

Reviewer 1

Absorbable metals or metallic stents, metals that corrode in physiological environment, constitute a new class of biomaterials intended for temporary medical implant applications. The results presented in the manuscript are new and interesting since Absorbable metals are suggested to use for ureteral stent applications. Moreover, new absorbable Zn-Al alloy also is suggested to use for this purposes. However some improvement of the manuscript should be done:

Introduction part. Besides of the Review article of H. Hermawan [12], two additional important Reviews also should be cited and discussed in the manuscript: (1) Recent Advances in Biodegradable Metals for Medical Sutures: A Critical Review; by J.M. Seitz et al. ADVANCED HEALTHCARE MATERIALS 4 (2015) 1915-1936 and (2) Biodegradable Metals for Cardiovascular Stents: from Clinical Concerns to Recent Zn-Alloys; by P.K. Bowen et al. ADVANCED HEALTHCARE MATERIALS 5 (2016) 1121-1140.

Both references are of high quality review published in top journals. We kept ref 12 to support the utilization of the term “absorbable” as mentioned in ASTM standards: F3160-16 and F3268-18, and we added (2) in the reference list. Thank you.

Figure 2 a and b. The data regarding Fe viability of urothelial cell and potentiodynamic polarization curve should be removed since they are redundant in this manuscript.

We preferred to keep the figures in the manuscript to maintain the flow of the narration.

Figure 6. The beginning of the caption for figure “Scanning electron microscope images...“ should be changed to “SEM micrographs…”.

Changed, thank you.

Reviewer 2

This manuscript describes the biological assessment of absorbable metals in view of ureteral stent application in view of ureteral stent application using normal human urothelial cells (NHUC) and metal direct. Overall, I think the work is satisfactory for publication with "Materials". Some minor comments are:

Are the authors sure that Mg(OH)2 is produced in the reaction of Mg and H2O? What conditions must be met for such a reaction to take place?

We thank a lot the reviewer for this comment. The brief sentence in page 9 of the manuscript has been revised and a supporting reference is cited. Please allow us the explain the following: In most aqueous environments, including the physiological environment used in this work, magnesium corrodes following these reactions:

Anodic reaction: Mg(s) → Mg2+(aq) + 2e, and

Cathodic reaction: 2H2O(aq) + 2e− → 2OH(aq) + H2(g), or

Overall reaction: Mg(s) + 2H2(aq) → Mg(OH)2(s) + H2(g)

Even though the formed hydroxide layer covers the surface of magnesium, this layer is not stable in the presence of chloride ions in the test solution. It is quickly converted into highly soluble magnesium chloride, through:

Mg(OH)2(s) + 2Cl(aq) → MgCl+ 2OH(aq)

The disappearance of the hydroxide layer hastens the corrosion of magnesium while the associated hydrogen gas evolution creates gas bubbles which causes the separation of tissues and/or tissue layers.

Refs:

Magnesium Implants: Prospects and Challenges, https://doi.org/10.3390/ma12010136 The effect of hydrogen gas evolution of magnesium implant on the postimplantation mortality of rats, https://doi.org/10.1016/j.jot.2015.08.003

The authors offer TEM images in Fig. 6, in my opinion is necessary to add an element of mapping for Zn and Mg.

Figure 6 shows some SEM images (not TEM) and it were taken with the interest of showing cell attachment, therefore we did not do any compositional mapping.

Reviewer 2 Report

This manuscript describes the biological assessment of absorbable metals in view of ureteral stent application in view of ureteral stent application using normal human urothelial cells (NHUC) and metal direct. Overall, I think the work is satisfactory for publication with "Materials". Some minor comments are:

Are the authors sure that Mg(OH)2 is produced in the reaction of Mg and H2O? What conditions must be met for such a reaction to take place? The authors offer TEM images in Fig. 6, in my opinion is necessary to add an element of mapping for Zn and Mg.

Author Response

(The authors gave the same response as above.)
